# Bertalanffy-Pütter models for avian growth

**Norbert Brunner**[ID][⊗], **Manfred Kühleitner**[ID]*[⊗], **Katharina Renner-Martin**[⊗]

Department of Integrative Biology and Biodiversity Research (DIBB), University of Natural Resources and Life Sciences (BOKU), Vienna, Austria

⊗ These authors contributed equally to this work.
* manfred.kuehleitner@boku.ac.at

## Abstract

This paper explores the ratio of the mass in the inflection point over asymptotic mass for 81 nestlings of blue tits and great tits from an urban parkland in Warsaw, Poland (growth data from literature). We computed the ratios using the Bertalanffy-Pütter model, because this model was more flexible with respect to the ratios than the traditional models. For them, there were a-priori restrictions on the possible range of the ratios. (Further, as the Bertalanffy-Pütter model generalizes the traditional models, its fit to the data was necessarily better.) For six birds there was no inflection point (we set the ratio to 0), for 19 birds the ratio was between 0 and 0.368 (lowest ratio attainable for the Richards model), for 48 birds it was above 0.5 (fixed ratio of logistic growth), and for the remaining eight birds it was in between; the maximal observed ratio was 0.835. With these ratios we were able to detect small variations in avian growth due to slight differences in the environment: Our results indicate that blue tits grew more slowly (had a lower ratio) in the presence of light pollution and modified impervious substrate, a finding that would not have been possible had we used traditional growth curve analysis.

**Data Availability Statement:** The data underlying the results presented in the study are available in the Supporting Information File S1 File.xlsx.

**Funding:** The author(s) received no specific funding for this work.

## Introduction

This paper uses nonlinear regression models to study the growth of passerine birds, blue tits (*Cyanistes caeruleus*) and great tits (*Parus major*). Our study was based on data from [1–4] on the development of 81 nestlings of blue tits and great tits, and the environmental characteristics of their nest sites (nest-boxes) in a public park of the Warsaw city center.

Ricklefs [5] was amongst the first to recommend simple regression models for exploring the growth patterns of nestlings. Examples include the S-shaped (sigmoidal) growth curves of the logistic trend model. It was used to relate growth to predation [6, 7], to environmental variation [8–10], to annual adult mortality rates, and to incubation period duration [11]. Literature has considered various environmental factors that may affect growth, such as: tree cover [12], impervious surface [4, 13], pollution by light [14, 15] and sound [16–18], or nest interference by humans and pets [1, 2, 19]. All these factors are also known to affect the breeding success. For example, when clearly distinct environments were compared (forests and urban parklands), then significant differences in the breeding success of blue tits [19] and in the growth of nestlings [10] could be established by the logistic and other simple models.

**Competing interests:** The authors have declared that no competing interests exist.

We aimed at demonstrating that less distinct environmental variation and the ongoing adaption of birds to urban environments [20] at a small scale could potentially be detected with more complex models. We used the five-parameter Bertalanffy-Pütter regression model (BP-model) for this purpose. It generalizes the traditional three-parameter regression models (e.g., Bertalanffy, Brody, Gompertz, Richards, or logistic growth of Verhulst) and provides a common biophysical interpretation for them (explained below). The growth function $m(t)$ of the BP-model describes mass ($m$) at time ($t$) using the following differential equation of Pütter [21].

$$m'(t) = p \cdot m(t)^a - q \cdot m(t)^b \tag{1}$$

The model parameters of Eq (1) are to be determined from fitting the model to mass-at-age data: Four parameters are displayed in the equation, namely the non-negative exponent-pair $a < b$ and the scaling constants $p$ and $q$. An additional parameter is the initial value (intuitively the hatching mass), i.e. $m(0) = c > 0$. This model was recently recommended in epidemiology [22–24] and it was also proposed [25] and applied [26–29] for studies in animal growth, where the BP-model (1) achieved significant improvements over the logistic model and other simpler three-parameter models with respect to the fit of the data. However, the benefits of its use remained little known and insufficiently evaluated. As explained below, the good fit was not the sole reason for using the BP-model: [30] argued that a better fit alone would not justify the added efforts of more complex models. Rather, we addressed a major concern about the traditional models, their inflexibility with respect to the shape of the growth curve [31, 32]. A main advantage of using the BP-model over three-parameter models is its flexibility due to two additional parameters (the variable exponent-pair) resulting in a larger pool of possible shapes of the growth curves.

For each nestling we identified the BP-model (its optimal parameters) with the best fit to the growth data. This approach avoided pooling across individuals (fitting growth curves to average-size-at-age). While pooling would be computationally easier (one fit for all females and males of each species) and the variations of the averages might be smaller (resulting in better fits), the aggregate data may miss biologically important differences between individuals (averaging treats them as equal). We used these models to compute the shape-parameter "ratio", mass at the inflection point over the asymptotic mass. Fig 1 illustrates the meaning of

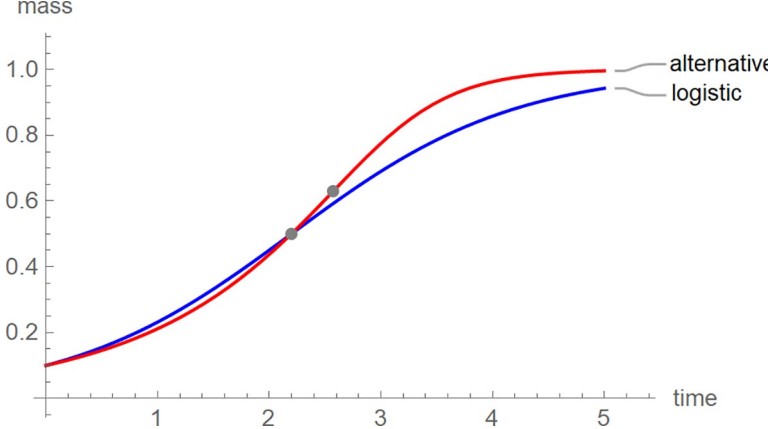

**Fig 1. Growth trajectories of logistic growth and an alternative growth model together with the inflection points.** The growth curves had the parameters a = 1, b = 2, c = 0.1, p = q = 1 (blue: logistic growth) and a = 1, b = 4, c = 0.1, p = q = 0.7526 (red). The asymptotic limit was 1 (both curves) and the inflection point coordinates (gray dots) were (2.197, 0.5) for logistic growth and (2.572, 0.63) for the alternative curve.

this parameter by the growth trajectories of two hypothetical birds. Although the initial and asymptotic masses were the same for both curves, the growth patterns differed: The growth curve with the higher ratio appeared to reach the asymptotic mass sooner (we refer to this situation as "faster growth"). We hypothesized that the shape of the growth curves might reflect the success of provisioning, which in turn may depend on the environmental situation. Therefore, we explored whether the shape-parameter ratio could reveal potential impacts of the environment on avian growth that could not be observed using logistic growth and other simple three-parameter models, whose ratios are fixed.

## Methods

### Nonlinear trend models

In temporal order the most common three-parameter growth models in avian research [33] are the Gompertz [34] model, the logistic model of Verhulst [35, 36], the monomolecular model (bounded exponential growth) of Brody [37], the von Bertalanffy [38, 39] model, and the more recent West model [40, 41]. There are also simpler trend models, such as power-laws between size and age [42], and more complex models explaining growth in relation to food consumption [43] or describing spatial characteristics of growth by partial differential equations [44].

The BP-model generalizes several common models in the following sense. If an exponent-pair $(a, b)$ is preset, then Eq (1) defines a model $BP(a, b)$ with three parameters ($c$, $p$ and $q$) as a special case of the general BP-model. The models of Bertalanffy and West are $BP(2/3, 1)$ and $BP(3/4, 1)$, respectively, monomolecular growth of Brody is $BP(0, 1)$, and logistic growth is $BP(1, 2)$. The Gompertz model is the limit case $BP(1, 1)$; see [45]. Common four-parameter models fit into this scheme, too: Richards [46] model is the BP-model with $a = 1$ (free parameters $b > 1$, $p$, $q$, and $c$); the "generalized Bertalanffy model" [39] is the BP-model with $b = 1$ (free parameters $0 \leq a < 1$, $p$, $q$, and $c$). Recently, the generalized logistic model [47] was recommended. It is represented by exponent-pairs of the form $(a, a+1)$. Assuming exponents $a = 1$ or $b = 1$, as for the above-mentioned models, then the differential Eq (1) could be solved by means of elementary functions [46, 48, 49]. For general exponent-pairs the solution of differential Eq (1) was expressed in terms of the Gaussian hypergeometric function $_2F_1$ [45, 50], which is not an elementary function [51].

[21, 39] interpreted Eq (1) as a model of ontogenetic growth, where the body would utilize resources at a metabolic rate ($p \cdot m^a$) for growth, except for the resources allocated to the operation and maintenance of existing tissue ($q \cdot m^b$). Different biophysical explanations for growth translated into different exponent-pairs ("metabolic exponents"). Thus, [38] proposed the exponents $a = 2/3$ as lung surface (assumed to be proportional to the $2/3^{rd}$ power of mass) delimits oxygen uptake, and $b = 1$ as cell count (assumed to be proportional to mass) determines the resources needed for maintenance. [40] developed a different argument in support of the exponents $a = 3/4$ and $b = 1$. This model was often used for mammalian growth [52] and [41] recommended it for avian growth. Other authors stressed that the metabolic exponents would also depend on the environment and not merely on biophysics [53, 54]. Hence, it would be meaningful to identify the best-fit exponent-pair for individuals.

### Shape-parameters

We use the term shape-parameter, but there is no need to formally define "shape" [55]. For solutions of Eq (1), the asymptotic mass $m_{max}$ and the mass $m_{infl}$ at the inflection point (peak

growth, reached at time $t_{infl}$) are computed from Eq (2):

$$m_{max} = \left(\frac{p}{q}\right)^{\frac{1}{b-a}}, \quad m_{infl} = \left(\frac{a}{b}\right)^{\frac{1}{b-a}} \cdot m_{max} \tag{2}$$

For $q = 0$ asymptotic mass is infinite and there is no inflection point (examples: linear growth $a = 0$, unbounded exponential growth $a = 1$). For $a = 0$ there is no inflection point, either (example: Brody model). For birds without inflection point we recorded $m_{infl} = 0$ and $t_{infl} = 0$. For the general model (1) the age $t_{infl}$ at the inflection point could only be determined numerically (solving $m(t) = m_{infl}$ for $t$). Note that an infinite (or unreasonably large) asymptotic mass did not always mean a poor fit of the growth curve to the data. Rather, the growth process might have been truly unbounded, as suggested for *Drosophila* larvae [39], or the data could have been representative for the initial (exponential) phase of growth, only. The parameters $m_{max}$, $m_{infl}$, and $t_{infl}$ are known to have a biological meaning. Asymptotic mass was linked to adult mass [56]. The inflection point was related to a change in diet, from spiders with much keratin (for the growth of feathers) to protein-rich caterpillars [57]. Further, the maximal growth rate, the derivative $m'(t_{infl})$, was suggested as a proxy for the basal metabolic rate [58].

This paper explores the shape-parameter ratio $m_{infl}/m_{max}$. As the ratio was defined from two biologically meaningful parameters, it might be biologically relevant, too.

## Bird and nest-box data

We use life history data from [1–3] and environmental data from [4] that were collected between March and June of 2016. [3] investigated the development of nestlings of blue tits and great tits that grew up at various sites in and around Warsaw, Poland. [4] complemented these growth data by reporting environmental data from the nest-box sites. We received from these authors data on 429 nestlings from 56 nest-boxes. For each bird, data included the ID of the bird (band number) and of the nest-box, the species (blue tits or great tits), the hatching date, the number of hatchlings at the first visit, mass in gram (eight repeated weighing), sex, and fledging success. To distinguish birds prior to banding, they were marked at the first measurement (using different methods, such as coloring or clipping of nails).

To ensure comparability, we studied only a fraction of the 429 nestlings. First, we selected only hatchlings from the first brood. Second, to exclude irregular growth patterns from runts, we selected those birds whose sex could be determined (female or male) and that finally fledged. Third, all selected birds were weighed at the same eight odd numbered days 1 (hatching), 3, 5, 7, . . ., and 15. [8] suggested to truncate the data at the peak, but we did not truncate our data as this could leave not enough points of time to determine the five model-parameters. Fourth, we focused on the study site with the largest number of surviving birds from different nests, Pole Mokotowskie park close to the city center (20 nest-boxes with 81 nestlings). Fifth, species and sex affected body mass (Fig 2), so we controlled for these factors: There were 26 female and 31 male blue tits and 12 female and 12 male great tits.

To study the association between environmental variables and growth we used the following information about the nest-boxes: human activity (explained in [1]), air temperature at the nest-box site (average of the morning temperatures in ˚C at the eight visits), light and sound pollution at the nest-box sites (units: lux and decibel), distances to paths and roads (minimal Euclidean distance in m from the nest-box to the center of the next path or road), percentage of impervious surface around the nest (ISA), tree cover in percent around the nest, and NDVI (normalized difference vegetation index). According to [3], the latter indicators were assessed

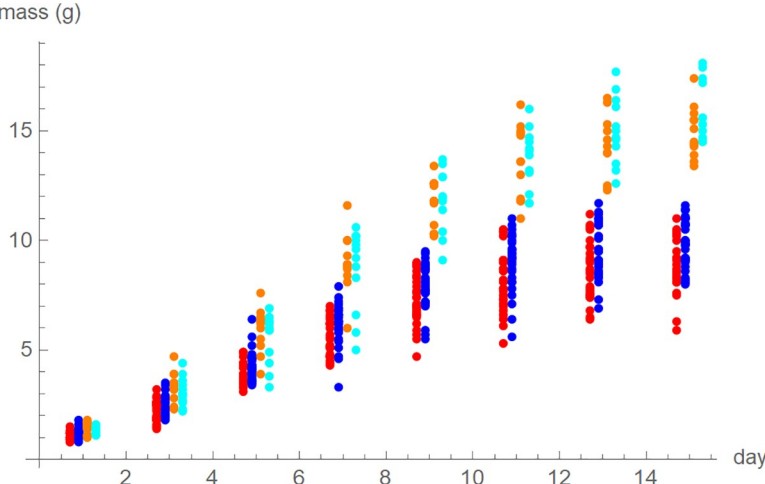

**Fig 2.** Mass at days 1, 3,. . ., and 15 plotted side-by-side; from left female blue tits (red), male blue tits (blue), female great tits (orange), and male great tits (cyan).

for a circle with radius 100 m around the nest and a pixel-resolution of 20 m. For methodological details we refer to the published sources [1–4].

In addition, we considered the association of growth to brood-specific indicators, namely the hatching day, the count of hatchlings at the first visit (4–12 birds/nest at day 1), at the last visit (1–9 birds/nest at day 15), and the difference between these counts. This difference may have been caused by early fledging, by death of runts, or by predation. In the latter case, the likely predators were woodpeckers, as they did not harvest all nestlings at once. Nests that were destroyed (typically, by martens) were removed from the sample.

## Calibration

There are different techniques to fit growth curves to data and to analyze them further. A large body of literature used the method of least squares and variants of it. Other approaches used spline-interpolation [59, 60], time series methods [61], or stochastic (partial) differential equations [62]. More recently, mixed-effect models became popular [63–65], but to remain viable they were based on growth curves that could be parameterized by means of elementary functions such as for the Verhulst model [66] or the Richards model [67].

We decided to use a variant of the method of least squares, sum of squared errors between log-transformed mass and log-transformed predictions (*SSLE*), because in previous work *SSLE* performed well for poultry and dinosaurs [27, 28]. Thus, given the growth data of a chick, we aimed at finding parameters that minimized *SSLE* of Eq (3). If $m(t)$ was a solution of Eq (1), using certain exponents $a < b$ and parameters $p, q, c$, and if $(t_i, m_i)$ were the mass-at-age data (for our data: $n = 8$ = number of measurements per chick), then *SSLE* was defined by Eq (3):

$$SSLE = \sum_{i=1}^{n} \left( \ln \left( m_i \right) - \ln \left( m(t_i) \right) \right)^2 \qquad (3)$$

To justify the logarithmic transformation, we checked for each bird, if its data $\ln(m_i)$ were normally distributed (lognormal distribution of body mass) and if the residuals in Eq (3) were normally distributed (*SSLE* as a maximum likelihood estimation). We used the Anderson & Darling [68] distribution-fit test to verify these assumptions.

$SSLE$ was related to a method of weighted least squares using $1/m(t_i)^2$ as weights [69], since its residual, $\ln(m_i)-\ln(m(t_i))$, approximated the relative error, $(m_i-m(t_i))/m(t_i)$. As [24] noted, the choice of the method calibration depends on the purpose of data-fitting. The logarithmic transformation of the data in Eq (3) was motivated by the Box & Cox [70] power transformations that aim at stabilizing variance, making data-fitting less sensitive to higher variances for the higher masses.

## Data-fitting

For the common three-parameter models there are various software solutions for determining the best-fitting parametric growth curves [71, 72]. However, literature reported difficulties with data-fitting already for four-parameter models, the generalized Bertalanffy model [73] and the Richards model [74]. To simplify data-fitting, several authors suggested to reduce the number of free parameters by not optimizing certain parameters. We did not apply this strategy. For instance, [43] defined the parameter $c = m(0)$ by the hatching mass (the first weighing at day 1 corresponded to the age $t = 0$), as this reduced the number of optimized parameters by one. In contrast, we used hatching mass as an estimate for the initial value but allowed the optimization to identify a better value of $c$ to find a lower $SSLE$. [56] recommended to identify asymptotic mass with the average adult age of the considered group of birds. We did not use this extrapolation, as after fledging and prior to attaining adult mass there might have occurred another growth phase, as was observed for farmed birds [75]. Thus, adult mass might overestimate the asymptotic mass of the considered growth phase.

To resolve these difficulties, for our data we simplified the parameter-space by using a grid-search combined with a custom-made variant of the method of simulated annealing [76] for data-fitting. Thereby, we defined a grid of possible exponent-pairs and for each grid-point exponent-pair $(a, b)$ we identified the best-fit growth curve for the three-parameter model $BP(a, b)$ by means of simulated annealing. Simulated annealing successively altered the current parameters slightly at random (we generated positive parameter-values $c, p, q > 0$, only) and compared the fit. Other than a random search (continuing with the better fit), simulated annealing with a positive probability allowed to continue with the poorer fit, whence the search could escape from suboptimal local extreme values. However, owing to the random character of this search procedure best fits were not always guaranteed. We therefore used a fine grid with distance 0.01 between neighboring grid-points, as then possible optimization errors at one grid-point could be corrected by the outcomes for the surrounding grid-points. Amongst all grid-point exponent-pairs (each representing a distinct three-parameter model) we then identified the one with the overall best fitting growth-curves. If the optimum was on the boundary of the grid, we manually added further grid-points. (For each bird, we optimized 37,454 to 98,903 grid-points, in the median 43,578.) We used Wolfram Mathematica 12 (www.wolfram.com) for our computations; for (explanations of) the Mathematica-code we refer to [27, 28].

## Model comparison

To assess the goodness of fit across different datasets (nestlings), we considered an analogue of the coefficient of determination for $SSLE$, namely RL-squared of Eq (4). It assesses the relative improvement of $SSLE$ for the best-fit BP-model in comparison to the best-fitting constant model (geometric mean of the masses):

$$RL^2 = 1 - \frac{SSLE(\text{BP model})}{SSLE(\text{constant})} \tag{4}$$

In view of certain limitations of R-squared [77] that apply also to $RL^2$, we did not draw further conclusions about model selection, as RL-squared would not be sufficiently selective for such a purpose. Instead, for model selection we used a variant of the Akaike information criterion (5), applied to the log-transformed data and models. It penalizes the model with more parameters, and the penalty is higher for fewer data [78, 79]. We used RL-squared merely to inform about the goodness of fit by means of a well-known statistic.

$$AIC_c = n \cdot \ln\left(\frac{SSLE}{n}\right) + 2 \cdot K + 2 \cdot K \cdot \frac{K+1}{n-K-1} \tag{5}$$

Here, $n = 8$ is the number of data-points for each bird, and $K$ is the number of optimized parameters of the model. ($K = 6$ for the general BP-model, counting $a, b, c, p, q$ and $SSE$, and $K = 4$ for logistic growth, where $a = 1$, $b = 2$ are not optimized.) When comparing two models, the model with the lower $AIC_c$ was selected.

Utilizing the information from optimization, we aimed at quantifying the possible consequences of overfitting. For each bird we identified those grid-point exponent-pairs $(a, b)$, where the corresponding three-parameter model, $BP(a, b)$, had a good fit to the data in terms of RL-squared (e.g.: $RL^2 \geq 95\%$ or $RL^2 \geq 99.5\%$). For each of these models we computed the parameter values of interest. We then evaluated the spread of parameters by means of the quantiles of the parameter-values that were computed for this set of models.

## Statistical approach for analyzing the ratio

We studied environmental indicators from avian literature that potentially affect the growth of birds. To find any potential environmental impact, we tested 13 indicators, but for concerns about spurious outcomes by chance we used three different approaches: a) correlation tests between the shape-parameter ratio and the indicators, b) location tests for the ratio of birds from clearly distinct environments (high vs. low indicator values), and c) location tests of the environmental parameters for birds with clearly distinct shapes of the growth curves (high ratio vs. low ratio birds). If the same association was 1.) supported by three different tests, 2.) was highly significant for at least one of them ($p \leq 0.01$), and 3.) was shared by the females and males of the same species or even by different species, then we deemed the outcome as reliable and reported it in detail. For completeness, the results mention other significant findings, too ($p \leq 0.05$).

We started with data-fitting that identified the best-fit BP-growth curve for each individual bird. For the further analysis of the shape-parameter ratio, we controlled for species and gender. This stratification defined four samples (female and male blue and great tits) of moderate sample sizes (12–31 individuals). To keep sample sizes large enough for statistically significant conclusions we did not control for nest sizes, but we used resampling techniques to explore this influence (see below.).

The search (a) for significant correlations between the shape-parameter (ratio) and the 13 (brood-specific and environmental) indicators applied the Spearman rho and the Spearman rank test [80] for each of the four samples. We expected at most two spurious outcomes per sample and test series. (P-value $p = 0.0245$ for three or more false reports of "95% significant", assuming a binomial distribution with 13 trials and a chance of 5% for errors.)

We then asked (b): Did variations in the local environments (at or around the nest-sites) lead to different shapes of the growth curves of the nestlings? Were nestlings from inferior environments inhibited in their growth? As thresholds between inferior and superior environments we used the medians of Table 1 for each indicator and group of birds. Thus, we divided each of the four samples into two groups of approximately equal sizes, namely birds that grew

**Table 1. Medians of the environmental data for each group of birds and counts of above/below median birds.**

| Indicator | Female blue tits | count[12] | Male blue tits | count[12] | Female great tits | Male great tits |
|---|---|---|---|---|---|---|
| Hatching day[1] | 38 d | A: 15, B: 11 | 38 d | A: 24, B: 7 | 36 d | 38 d |
| Hatchlings initially[2] | 9 | A: 20, B: 6 | 7 | A: 25, B: 6 | 7 | 7 |
| Hatchlings finally[2] | 6 | A: 15, B: 11 | 5 | A: 17, B: 14 | 5 | 4 |
| Nest-size difference[2] | 5 | A: 17, B: 9 | 1 | A: 23, B: 8 | 1 | 1 |
| Human activity[3] | 1.1 | A: 15, B: 11 | 1.05 | A: 20, B: 11 | 1.1 | 0.75 |
| ISA[4] | 4.59% | A: 13, B: 13 | 4.49% | A: 20, B: 11 | 10.38% | 9.07% |
| Light[5] | 3160 lx | A: 14, B: 12 | 2852 lx | A: 20, B: 11 | 8588 lx | 4907 lx |
| NDVI[6] | 0.76 | A: 14, B: 12 | 0.75 | A: 17, B: 14 | 0.69 | 0.71 |
| Path[7] | 14.3 m | A: 11, B: 15 | 11.1 m | A: 17, B: 14 | 5.55 m | 6.9 m |
| Road[8] | 85.4 m | A: 15, B: 11 | 165.1 m | A: 18, B: 13 | 63.7 m | 106.9 m |
| Sound[9] | 65.44 db | A: 13, B: 13 | 65.93 db | A: 16, B: 15 | 65.41 db | 66.98 db |
| Temperature[10] | 10.86˚C | A: 17, B: 9 | 10.86˚C | A: 20, B: 11 | 11.04˚C | 11.08˚C |
| Tree cover[11] | 28.13% | A: 14, B: 12 | 19.07% | A: 16, B: 15 | 30.46% | 24.90% |

**Notes**: All numbers were rounded to the shown decimal.

[1] count of days from the start of the field work to hatching

[2] initial and final counts, respectively, of hatchlings in each nest, and difference of these counts

[3] index of human activity from Corsini et al. (2017)

[4] percentage of impervious area around the nest-box

[5] light pollution at the nest-box site (in lx)

[6] index for the vegetation cover around the nest-box

[7] minimal (Euclidean) distance to the (center of the) next path (in m)

[8] minimal (Euclidean) distance from the nest-box to (the center of) the next road (in m)

[9] sound pollution at the nest-box site (in db)

[10] average of the morning temperatures at the eight visits (in ˚C)

[11] tree cover around the nest-box (percent)

[12] count of birds of the given type (first row, left column), whose nest characteristics was above (A)/below (B) the median (displayed in the left column). Counts for great tits are not displayed, because for 12 birds per group the subsamples were too small to be representative.

up in an ambience with a high parameter value (equal or larger than the sample median) and the other birds. For these classes we tested for significant differences in the location of the ratio using the Mann-Whitney test [80].

We finally explored (c), if and to what extent the growth-patterns allowed to draw inferences about the "social background" of birds: Did birds with distinct ratios come from different environments? We defined subsamples of the four groups of birds in terms of the ratio: A bird was high-ratio, if $m_{infl}/m_{max} \geq 0.5$ for the best-fit BP-model growth-curve (the threshold 0.5 comes from logistic growth), and otherwise it was low-ratio. Using the Mann-Whitney test we compared the medians of the environmental indicators for the subsamples of high-ratio and low-ratio birds.

Finally, we checked for possible bias. Owing to the correlations between the environmental and nest-specific indicators, in theory the count of spurious observations for the ratio could be larger than initially estimated. To obtain more precise estimates we used resampling: For each group of birds we reshuffled the birds at random amongst the nest-sites (the nest characteristics were not altered) and counted the so obtained supposedly false significant associations. Our improved estimate was the 95%-quantile of these counts from of 10,000 shuffles. (The high number of simulations was motivated by a recommendation [81] for Mantel's test.) Further, we explored, if the outcomes were affected by nest-size. Assuming that growth was

affected by the ability of the parents to supply each nestling with adequate food, then this ability was influenced by nest-size and not only by environmental factors. We used resampling to reduce such dependencies: For each group of birds we selected one "representative" bird per nest and repeated the previous analysis of correlations for this small sample.

## Results

### Best-fit exponent-pairs and goodness of fit

We first observed that the logarithmic transformation stabilized variance. This confirmed the suitability of *SSLE*. For each bird we tested if its masses ($m_i$) were log-normally distributed. The Anderson-Darling test did refute this assumption for six (7.4% of 81) birds (P-values $p < 0.05$). Further, the *SSLE* residuals were significantly non-normal in only 6.2% of chicks (5 of 81), and for three nestlings both distribution assumptions were refuted. We did not discard of these data, as these were only few exceptional birds.

The best-fit exponent-pairs appeared to be scattered at random, whereby the exponent $b$ was much more variable than the exponent $a$ (Fig 3). We could not observe a concentration near any of the exponent-pairs of the "traditional" three-parameter models (Brody, Bertalanffy, Gompertz, West, Verhulst). For six birds the exponent-pairs were on the line $a = 0$ (no inflection point). 47% of the birds were near-diagonal (exponent-difference smaller than 1.5), whereby the proportion of near-diagonal birds was almost equal amongst all groups of birds: 50%, 48%, 42%, and 42% for female and male blue tits, and female and male great tits, respectively.

Next, for each bird we screened the growth curve of the BP-model with optimized parameters to assess its fit to the data. In general, the five-parameter BP-model achieved an excellent fit with median-$RL^2$ of 99.77%; for 70 of 81 data RL-squared was above 99.5%. However, one bird was exceptional (K7V3278), as the best-fit BP-model did not achieve RL-squared above 95% (namely $RL^2$ = 94.2%). For this nestling, asymptotic mass was excessive ($1.4 \cdot 10^{13}$). This was also insofar exceptional, as generally asymptotic mass was close to the maximal observed mass: The median of the quotient of the asymptotic mass over the maximal observed mass was 1.006 with the 95% confidence interval between 0.998 and 1.016 (computed from one-sided sign tests [79] with $p = 0.0224$ for both limits). Further, we evaluated the maximal absolute deviations of the model curves from the data. For blue tits, in the median the maximal deviation was 0.41 g; worst case: 0.91 g for the exceptional bird (day-15 mass: 9.8 g). For the larger

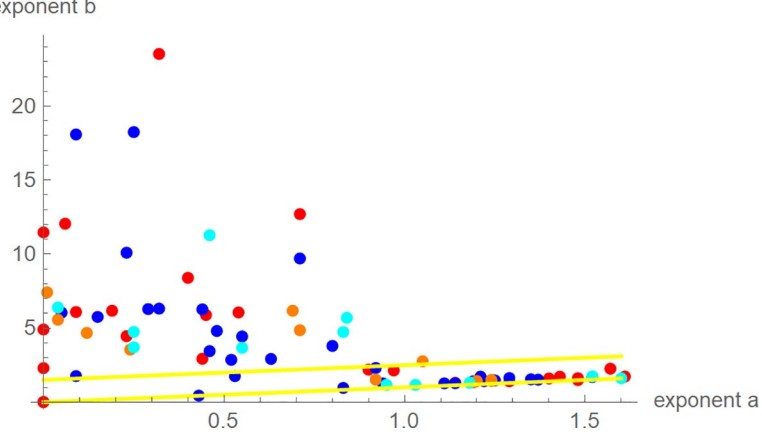

**Fig 3.** Best-fit exponent-pairs for 81 nestlings of female blue tits (red), male blue tits (blue), female great tits (orange), male great tits (cyan), and yellow lines (b = a and b = a+1.5) to indicate near-diagonal exponent-pairs.

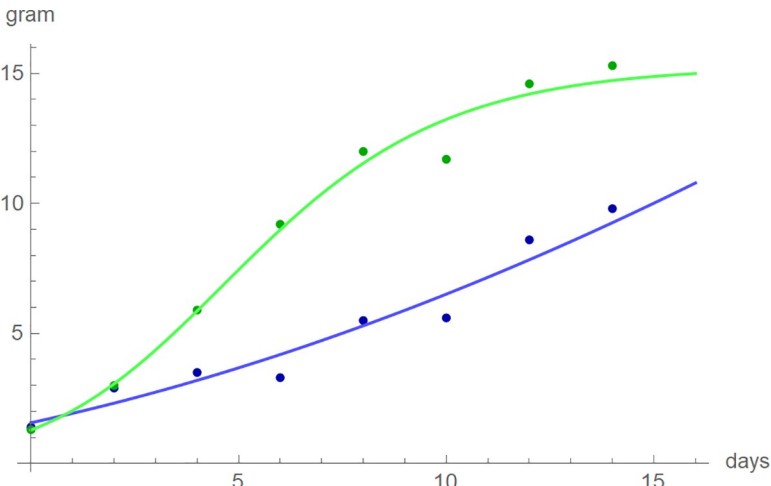

**Fig 4. Mass and best-fitting BP-model growth curves of blue tit K7V3620 (blue) and great tit K7V3278 (green), chosen for their poor fits.** (Days 1, 3,. . ., and 15 correspond to t = 0, 1,. . ., 14.).

great tits, in the median the maximal deviation was 0.64 g; worst case: 1.53 g for chick K7V3620 (day-15 mass: 15.3 g). Fig 4 plots the growth data of these two birds together with the best-fit model curves. Apparently, the deviations were caused by fluctuations of the growth data, while the model curves captured the correct shapes of the growth data. As such fluctuations are common [82], we did not exclude any birds.

Model selection per se was not the primary issue of this paper, because for conceptual reasons (flexible ratio $m_{infl}/m_{max}$) we decided to use the five-parameter BP-model, even if there was a danger of overfitting (meaning that simple models might have been more parsimonious). Indeed, when compared with the best-fit BP-model, logistic growth had a lower $AIC_c$ for all birds. The reason was the huge penalty for the BP-model from the last term in Eq (5). The generalized logistic model [47] might be a more parsimonious four-parameter alternative to the general BP-model. For this model, $AIC_c$ was always lower than for the general BP-model, but still $AIC_c$ was higher than for logistic growth, even for the near-diagonal birds.

### Ratios for best and nearly best fitting growth-curves

In view of Eq (2) the ratio, $m_{infl}/m_{max}$, depends on the exponent-pair, only. For any given three-parameter model $BP(a, b)$ with a preset exponent-pair, the ratio is fixed, irrespective of the data (e.g. ratio 0.5 for logistic growth, unless $q = 0$). For the general five-parameter BP-model and for the four-parameter generalized logistic growth model any ratio between 0 and 1 could be realized. By contrast, for the four-parameter Richards-model the ratio was bounded from below by $1/e$ (= 0.368) and for the generalized Bertalanffy model it was bounded from above by $1/e$. For our data, the best-fit ratios ranged from 0 (for $a = 0$) to 0.835, whereby 0.1 was the least ratio for a sigmoidal bird (meaning $a > 0$). For 21 birds (19 of them sigmoidal) the ratio was smaller than $1/e \approx 0.368$ (lower bound for the Richards model) and for 48 birds it was above 0.5 (ratio of logistic growth).

Bertalanffy [39] mentioned the ratio and suggested that its value would be around 1/3 for vertebrates. Was this true for our data? Comparing the ratios of sigmoidal birds with the value 1/3, for all four groups of birds the ratios were significantly greater than 1/3 (one-sided sign test [80]: $p < 0.015$ for each group). Rather, the median ratios (with 95% confidence bounds) were close to the ratio 0.5 of logistic growth: 0.511 (0.467–0.59) for female blue tits, 0.503

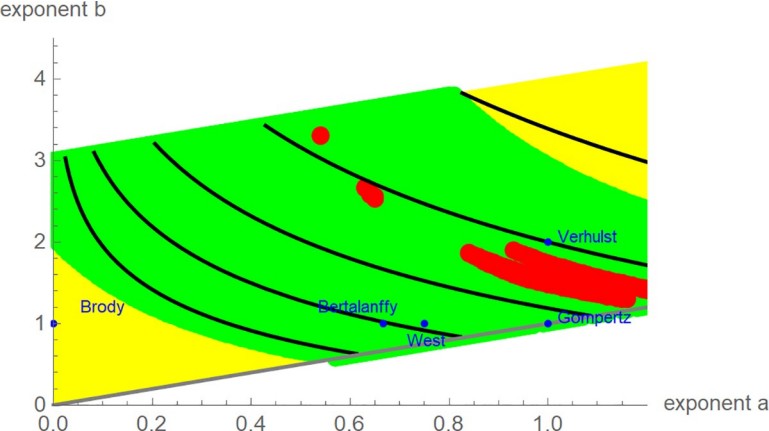

**Fig 5.** Exponent-pairs of named models (blue), exponent-pairs of models with RL > 99.5% for 66 data (red), exponent-pairs of models with RL > 95% for 80 data (green), remaining exponent-pairs of the search grid (yellow), diagonal a = b (grey) and contour lines (thick and black) of the ratios 0.2, 0.3,. . ., 0.6 (from left to right), using Eq (2) for the ratio.

(0.467–0.57) for male blue tits, 0.459 (0.42–0.59) for female great tits, and 0.528 (0.42–0.65) for male great tits.

To study the variability of the model parameters, we identified for each bird (except K7V3278) the exponent-pairs that defined BP-models with a good fit: $RL^2 \geq 95\%$. Given any of these 80 birds, there was a large region of exponent-pairs that represented alternative models, whose growth curves had a good fit to the growth-data and that therefore had about the same shape: The green area in Fig 5 plots the exponent-pairs that were common to all these regions. The best-fit ratio turned out to be insofar resilient against this variability, as the median of all estimated values from growth curves with excellent fit was often close to the best-fit value. To give a specific example, there were 66 birds with sigmoidal best-fit models satisfying $RL^2 \geq 99.5\%$. (The red area in Fig 5 plots the still large region of exponent-pairs that achieved an excellent fit for all 66 birds.) For 47 (≡ 71% of these 66) birds, the relative error between the best-fit ratio and the median of the ratios of all growth curves with $RL^2 \geq 99.5\%$ was below 10%. Similarly, for 47 birds the absolute difference between the best-fit ratio and that median was below 0.05.

## Statistical analysis of the ratio

We searched systematically for significant associations between the ratio, four brood-specific indicators, and nine environmental indicators, whereby we studied four groups of birds independently (female/male blue/great tits). Table 2 summarizes the main finding: The indicators impervious area (ISA) and light pollution had highly significant negative correlations with the ratio for both female and male blue tits. The strengths of these correlations were moderate to strong (using a classification in [83]). Further, nestlings of blue tits that grew up in environments with high percentage of impervious area (ISA) or a high level of light pollution had significantly lower ratios (highly significant in one case). And for high-ratio blue tits the light pollution and ISA was significantly lower than for low-ratio birds (but not highly significant). Thus, both for female and for male blue tits the associations between the ratio and the two indicators (ISA and light pollution) were supported by three different tests, some of them highly significant, whence we deemed them as reliable. The correlation tests achieved higher levels of significance, because they were applied to the full samples (26–31 birds), while the

**Table 2. Tests for significant associations between the ratios for blue tits and the indicators impervious area (ISA) and light pollution.**

| | | females | | males | |
|---|---|---|---|---|---|
| **Count n =** | | **26** | | **31** | |
| **Indicators:** | | ISA | light | ISA | light |
| *Correlation between ratio and indicators* | | | | | |
| Spearman rho | | **−0.54** | **−0.64** | **−0.48** | **−0.51** |
| P-value (Spearman rank correlation test) | | **0.004** | **0.0004** | **0.0069** | **0.0035** |
| *Comparing growth patterns from distinct environments* | | | | | |
| Median ratio: birds from nests with | high indicator value | 0.467 | 0.493 | **0.473** | 0.491 |
| | low indicator value | 0.529 | 0.576 | **0.595** | 0,599 |
| P-value (Mann-Whitney test) | | 0.0349 | 0.0415 | **0.0032** | 0.0157 |
| *Comparing environments for distinct growth patterns* | | | | | |
| Median indicator for | high ratio birds | 4.54% | 2896 lx | 3.58% | **2852 lx** |
| | low ratio birds | 13.33% | 5937 lx | 8.88% | **3160 lx** |
| P-value (Mann-Whitney test) | | 0.018 | 0.041 | 0.0114 | **0.002** |

**Note**: All mentioned outcomes were significant ($p < 0.05$); highly significant outcomes ($p < 0.01$) were displayed in boldface.

other two (location equivalence) tests compared smaller subsamples (e.g.: subsamples of 11–19 high/low ratio birds).

For blue tits we observed additional significant outcomes: Both for female and for male blue tits there was a highly significant negative correlation of the ratio with the difference between the initial and final nest sizes (median differences 3 and 6 for female high-ratio and low-ratio birds, respectively). For female blue tits, but not for males, there was a highly significant negative correlation of the ratio with initial nest-size (in the median high-ratio and low-ratio birds had 8 and 9.5 siblings, respectively), and there was a highly significant difference in the level of sound pollution between high-ratio birds (median 64 db) and low-ratio birds (median 68 db), which translated into a significant negative correlation. Further, for male blue tits, only, there was a significantly positive correlation of ratio with NDVI (and the ratios of birds from nests with high or low NDVI differed significantly).

For great tits, the sample size turned out to be too small. There were highly significant negative Spearman rank correlations of the ratio with the initial nest sizes for female and male great tits. The Mann-Whitney tests involved subsamples of merely 3–9 birds, which was hardly representative, so we did not analyze great tits further.

## Check for possible bias

The purpose of this paper was to illustrate the ability of the ratio (of BP-models) to detect biological effects. For male and female blue tits 4 and 5 of the 13 correlation tests between ratio and environmental or nest-specific indicators were significant, respectively, and at most two tests of each series were spurious by chance. Therefore, we could conclude that there were valid significant statistical associations between some environmental variables and the ratio.

However, our estimate for the count of spurious correlations assumed independence of the tests. For our data, the environmental indicators were correlated, whence the estimated count of spurious correlations could be overly optimistic. For instance, for the data of female and of male blue tits nest-size difference was significantly correlated with initial nest-size, final nest-size, hatching day, ISA, sound pollution, and temperature. In addition, for female blue tits there were significant correlations with light pollution and distance to a path, while for male blue tits there were significant correlations with human activity and NDVI. Several of these

correlations were strong and highly significant. To account for the correlations between the environmental and nest-specific indicators we used resampling. For 95% of 10,000 random shuffles of female or male blue tits amongst the nest there were at most two spurious correlations, confirming the previous estimate. (For female great tits up to three spurious correlations were conceivable.)

The highly significant correlations of the ratio with initial nest size for three groups of birds gave rise to the question if birds from large broods had dominated our observations: Were the significant associations of ratio with ISA and light pollutions for blue tits mere artefacts of the experimental design? We used resampling, selecting for each group of birds one representative nestling from each brood, to show that this was not the case. If for each nest we deterministically selected the bird (of the required gender) with "typical growth" (ratio nearest to the nest-median), then for both groups of female and male blue tits the ratio had a significant negative correlation with light pollution and ISA. (In view of the small sample sizes of 11 male and female birds, each, we could not expect a high significance. However, P-values were below 0.03.) Using bootstrap simulations (selecting the representative bird at random) confirmed this finding.

## Discussion and conclusions

We studied a shape parameter for bounded sigmoidal growth curves, the ratio of inflection point mass over asymptotic mass (range between 0 and 1), whereby we used the five-parameter BP-model to estimate the ratio. Using data from literature, we illustrated the utility of this model by demonstrating that a higher level of urbanization (impervious area, light pollution) was associated to a lower ratio, which in turn indicated a (finally) slower growth towards the adult (asymptotic) mass. These findings were confirmed by three different tests and they remained stable, when we eliminated the possible dependency on nest-size by resampling with one bird from each nest.

We conclude, irrespective of the biological interpretations of our results, that the ratio was suitable to detect small variations in avian growth due to fine-scaled environmental and brood-specific factors. However, we did not aim at pinpointing exact causations for the observed effects. Thus, the finding for ISA was biologically plausible, as impervious area around the nest could inhibit the success of provisioning [13] and this clearly would affect the growth of the nestlings. Light pollution, on the other hand, is known to affect the behavior of birds [14, 15], but from the data we could not discern a mechanism explaining its possible impact on the growth of nestlings. As for another example, sound pollution is known to affect the growth of certain passerine species adversely, as for instance in noisy environments female sparrows reduced the provisioning rate for their brood [18], but for our data the evidence for an impact of sound pollution on the growth of the nestlings was weak.

The count of $n = 8$ observations per bird, as for our data, was the minimum for a meaningful model comparison by means of $AIC_c$ of Eq (5). This small number of time-points used was also a reason for the high variability of the exponent-pairs with a good fit (Fig 5). More measurements per bird would be preferable and for our data the variability could have been reduced by weighing the nestlings daily instead of at every second day. For this reason, altricial birds that leave their nests late would be a good choice for growth studies (e.g., hawks), but many species are sensitive to nest interferences by investigators, breed at remote places, and are protected by the law. Passerine birds accepting nest-boxes are less problematic in this respect. For precocial birds our approach to growth modeling would require domestic animals (e.g., poultry [27, 59]) with environmental variation simulated by the experimental design. Many semi-precocial and semi-altricial, ground-nesting seabird chicks are ideal subjects for this type of study, too, since they can be recaptured successfully until they fledge [72].

A drawback of the BP-model was the excessive CPU-time needed for data fitting (up to a week per bird). However, the ratio is meaningful for any model, so researchers interested in studying the relation of the ratio to environmental indicators are not confined to the BP-model. While the traditional three-parameter models are not suitable for this task (inflexibility with respect to the ratio), there are suitable four-parameter BP-models, such as the generalized logistic model [47]. Fig 5 explains the reason: The ratios were constant on the black curves that were (approximately) perpendicular to the diagonal, and the line $b = a+1$ of exponent-pairs of the generalized logistic model was parallel to the diagonal, intersecting all these curves. For data-fitting, the five-parameter BP-model required the optimization of the model parameters for tens of thousands grid-point exponent-pairs, while for generalized logistic growth by the same procedure the optimization at a few hundred grid-point exponent-pairs on the line $b = a +1$ would suffice. Thus, using this four-parameter model would reduce CPU-time for computing the ratio by a factor of 100 to less than an hour per bird. (Using more data per bird, as recommended previously to reduce the variability, would slow down the computations slightly.)

We nevertheless used the five-parameter BP-model for our research, as initially we were interested in another question: We observed that for about half of the birds the best-fit exponent-pair was close to the diagonal (Fig 3). We had similar observations also for other species [29]. Was there a biological reason for this? This question was our starting point and we initially studied the exponent-difference $b – a$. However, for blue tits, the ratio was found to be more sensitive to biological signals, and for great tits the sample size (12 individuals of each gender) was too small for reliable findings about the exponent-difference.

We conclude that complex models, even if not parsimonious, allow to recognize interesting geometric patterns in avian growth (as with the ratio for blue tits). However, to identify parameters that quantify differences in growth patterns and that are not distorted by high variability, sufficiently large samples in terms of both birds and mass-at-age data are needed.

## Supporting information

**S1 File. Excel file with explanations of the columns in rows 1–2.** Columns A-D provide the bird IDs together with general information, E-Q provide the brood-specific and environmental data for each nestling (and nest), R-Y list the bird-mass at days 1 to 15 (odd days, only), and AA-AH provide for each bird the best-fit parameters of its five-parameter BP-model together with additional information about the model.
(XLSX)

## Acknowledgments

This paper originated from conversations with Michela Corsini and Eva Maria Schöll, who explained to us the methodology of the papers [1–4]. The data for these papers were collected under projects directed by Marta Szulkin as the principal investigator. She gave us the permission to publish the data as a supporting information (part of S1 File) for the present paper. As this research was ethically approved, there occurred no ethical issues for this paper.

The authors are grateful to Michela Corsini and Júlio Neto, who provided insightful comments on preliminary versions of our paper. Further, the authors appreciate the extensive comments by Steve Oswald and Tales J. Fernandes, the two reviewers of this journal. They helped to improve the paper substantially.

## Author Contributions

**Conceptualization:** Norbert Brunner, Manfred Kühleitner, Katharina Renner-Martin.

**Data curation:** Norbert Brunner, Manfred Kühleitner, Katharina Renner-Martin.

**Formal analysis:** Norbert Brunner, Manfred Kühleitner.

**Investigation:** Katharina Renner-Martin.

**Methodology:** Norbert Brunner, Manfred Kühleitner, Katharina Renner-Martin.

**Software:** Norbert Brunner, Manfred Kühleitner, Katharina Renner-Martin.

**Supervision:** Norbert Brunner, Manfred Kühleitner, Katharina Renner-Martin.

**Validation:** Norbert Brunner, Manfred Kühleitner, Katharina Renner-Martin.

**Visualization:** Norbert Brunner, Manfred Kühleitner, Katharina Renner-Martin.

**Writing – original draft:** Norbert Brunner, Katharina Renner-Martin.

**Writing – review & editing:** Norbert Brunner, Manfred Kühleitner, Katharina Renner-Martin.

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
