## [Decision Letter · Decision Letter 0]

10 Dec 2020

PONE-D-20-31677

Bertalanffy-Pütter models for avian growth

PLOS ONE

Dear Dr. Kühleitner,

Thank you for submitting your manuscript to PLOS ONE. After careful consideration, we feel that it has merit but does not fully meet PLOS ONE’s publication criteria as it currently stands. Therefore, we invite you to submit a revised version of the manuscript that addresses the points raised during the review process.

We look forward to receiving your revised manuscript.

Kind regards,

Dragan Pamucar

Academic Editor

PLOS ONE

Journal Requirements:

2. Please amend the manuscript submission data (via Edit Submission) to include author Katharina Renner-Martin.

3. We note that Figure S1 n your submission contains satellite images which may be copyrighted.

a. You may seek permission from the original copyright holder of Figure S1 to publish the content specifically under the CC BY 4.0 license. 

Reviewers' comments:

Reviewer's Responses to Questions

**Comments to the Author**

1. Is the manuscript technically sound, and do the data support the conclusions?

Reviewer #1: Partly

Reviewer #2: Yes

2. Has the statistical analysis been performed appropriately and rigorously? 

Reviewer #1: No

Reviewer #2: No

3. Have the authors made all data underlying the findings in their manuscript fully available?

Reviewer #1: No

Reviewer #2: Yes

4. Is the manuscript presented in an intelligible fashion and written in standard English?

Reviewer #1: Yes

Reviewer #2: Yes

5. Review Comments to the Author

Reviewer #1: See my review attached: "PONE-D-20-31677 Review_SAO.pdf"

Also, I have provided separate minor suggestions on the ms in the attached version of the ms: "PONE-D-20-31677_reviewer.pdf"

.................................................................

Reviewer #2: Review: Bertalanffy-Pütter models for avian growth

Very informative and didactic article on the subject of intense discussion in literature.

My recommendation is that it be published, provided that the discussion with another

papers in literature is improved.

Below are some suggestions to improve the presentation of the article.

Introduction

Take care to define the acronym the first time it appears.

For example, second paragraph, define the acronym BT.

In the fifth paragraph (page 4), you can add the discussion with the article

Fernandes et al. (2019) which also evaluated the allometric values (2/3, 1)

and (3/4, 1), concluding that for mammals it is better use (3/4, 1).

Sixth paragraph (page 4), Fernandes et al. (2020) also studied growth and solved the

Bertalanffy-Putter differential equation, including adding the relationship between

the "original" parameters with the three-parameter models. It is advisable to read

this paper to assist in the discussion and presentation of the results.

Computations and statistics

Statistical analyzes need to be better described. For example, to present the

statistics of the used tests.

Was the residue analysis performed after obtaining the estimates? Remember that this

is the most important step in studies with regression models. Present (or even comment)

on assumptions analysis

Authors should also assess the presence of heteroscedasticity. Looking at Figure 2,

it seems quite plausible that the variances between the weighing days are

heterogeneous (increasing over time).

Data fitting

Why were used the logarithm of data? This transformation interferes in the parameter estimates (Table 2).

Was the inverse transformation done, to return with the parameters in the original scale?

For example, was "b-a" calculated on the original scale or on the logarithmic scale?

Best fit model

Even justifying that the focus is not on comparing models, it could present the value of at least R

squared (R²), for the reader to have an idea of the quality of the adjustment obtained.

Compreendi que o foco do artigo está em outros parâmetros, mas nada impede de discutir também os parâmetros

de informações práticas, como peso adulto e abscissa (em dias) do ponto de inflexão. Veja a relação

indicada por Fernandes et al. (2020).

Discussion and Conclusions

The conclusion of the article is based on the exponential difference parameters b-a and the Minfl/Mmax ratio.

Therefore, they need to be better explained in the introduction. What is its biological importance?

Why is it important for the reader to be concerned with these parameters and not simply

adjust a three-parameter model?

Its results need to be discussed with articles already published in the literature

about the subject. Otherwise it is not an discussion.

Here are some suggestions for papers that can help in the discussion.

Fernandes et al. (2020)

https://doi.org/10.28951/rbb.v38i3.457

Fernandes et al. (2019)

https://doi.org/10.1590/S1678-3921.pab2019.v54.01162

6. PLOS authors have the option to publish the peer review history of their article (what does this mean?). If published, this will include your full peer review and any attached files.

Reviewer #1: **Yes: **Stephen A. Oswald

Reviewer #2: **Yes: **Tales Jesus Fernandes

---

## [Author Response · Author response to Decision Letter 0]

16 Feb 2021

See the attached File: 

Response_to_Reviews_4

---

## [Decision Letter · Decision Letter 1]

4 Mar 2021

PONE-D-20-31677R1

Bertalanffy-Pütter models for avian growth

PLOS ONE

Dear Dr. Kühleitner,

Thank you for submitting your manuscript to PLOS ONE. After careful consideration, we feel that it has merit but does not fully meet PLOS ONE’s publication criteria as it currently stands. Therefore, we invite you to submit a revised version of the manuscript that addresses the points raised during the review process.

We look forward to receiving your revised manuscript.

Kind regards,

Dragan Pamucar

Academic Editor

PLOS ONE

Journal Requirements:

Reviewers' comments:

Reviewer's Responses to Questions

**Comments to the Author**

1. If the authors have adequately addressed your comments raised in a previous round of review and you feel that this manuscript is now acceptable for publication, you may indicate that here to bypass the “Comments to the Author” section, enter your conflict of interest statement in the “Confidential to Editor” section, and submit your "Accept" recommendation.

Reviewer #1: (No Response)

Reviewer #2: All comments have been addressed

2. Is the manuscript technically sound, and do the data support the conclusions?

Reviewer #1: Yes

Reviewer #2: Yes

3. Has the statistical analysis been performed appropriately and rigorously? 

Reviewer #1: Yes

Reviewer #2: Yes

4. Have the authors made all data underlying the findings in their manuscript fully available?

Reviewer #1: Yes

Reviewer #2: Yes

5. Is the manuscript presented in an intelligible fashion and written in standard English?

Reviewer #1: No

Reviewer #2: Yes

6. Review Comments to the Author

Reviewer #1: I applaud the authors for their revisions. These really help bring out the impact of the study.

Although there have been many excellent language revisions, I still believe that the language can be more precise and briefer in many places. Since PLoS ONE does not copy edit following acceptance, I have taken the liberty to provide very detailed language suggestions throughout on a marked-up copy of the ms that I have uploaded here. Feel free to use these as you see fit.

There are still a few issues I have with the ms but these are with presentation not content (the authors have done a great job of toning down the ms in areas where the original became speculative). The main problem is that the end of Methods and Results are too long and arduous and the Discussion is too short. Since much of this is wording/formatting I have provided detailed comments on this below:

1) METHODS/RESULTS ARE VERY LONG. The following sections of the Methods and Results are unnecessary long and deter the reader:

a. Remove lines 129-136: this has already been covered adequately on lines 84-93.

b. Lines 143-151: This paragraph is a bit confusing in a methods section as it is talking about your results. I suggest two possible solutions: either Talk about it in more pilot-study or theoretical terms. e.g. "If we compare two BP-growth curves...." or move it to the Discussion. I also suggest moving this paragraph up to be before the line starting "This paper explores.." (Lines 127)

c. Lines 213-220: suggest move this entire paragraph to Supplementary Info since has already been used in other published studies

d. Lines 221-223: Since this is a single example, it would be good to show more examples for several reasons: 1) to give the reader more confidence that the fit is good across all individuals and that the error shown is consistent - perhaps quote the range of error across all study birds; 2) to show that the curve does better for shapes that are quite different from standard (the example shown looks visually a lot like a logistic curve). I believe that Fig 1 could be eliminated (since it is clear in the text that a flexible inflection point is what you are using) and Fig 3 could be expanded to include several difference examples.

e. Line 234-238: a bit tangental to the study in hand. It is reasonable to expect fitting to fail for 5 parameters on 8 datapoints – suggest removal

f. Line 277: since this is so high (43,000 models), there needs to be some indication of why - were exponent-pairs assessed to the nearest three significant figures? Or does this include models used to explore the environmental factors. How many possible models were there, how big was the grid-space?

g. Line 279-284: this is unusual phrasing because usually you describe which tests you did on which data, rather than what appears to be more of a philosophy behind the statistical approach. Since specific tests were matched to the specific analyses that used them below, I'd delete this paragraph

h. Lines 301-337. This needs substantial reduction and rephrasing. On reading the first paragraph of the Discussion, I finally understood the approach. I would suggest moving the first paragraph of the Discussion here, and just expanding it by putting in the bare necessary details from the existing method text. Shorter the better.

i. Lines 353-359: This is a good initial approach for when first examining the data. However, since it is one bird, I don't think it should take up a whole paragraph of the Results of the paper. Instead, justify it by saying it captured the correct shape and was only deficient in the height of the asymptote so you didn't exclude it.

2) RESULTS ARE HARD TO FOLLOW. The sections: “Identification of the ratio from the shape”, “Correlations with the ratio”, “Further associations with the ratio for blue tits” and “Related Issue” need to be shortened and renamed. I would suggest signposting from the analysis roadmap provided in response to my comment “h” above by giving easier to follow names that represent each of the 3 analytical approaches used [a) correlation tests, b) location tests for the shape-parameter (ratio) of birds from clearly distinct environments, and c) location tests of the environmental parameters for birds with clearly distinct shapes of the growth curves]. Also, substantial deletions to each of these sections is necessary to avoid losing the reader in unnecessary detail:

a. Lines 387-422. This is very long and unclear. On the attached marked-up ms, I have provided considerable edits to make this brief and to-the-point. Please address this.

b. Lines 385-386, 430-434. These are Methods and should not be in the Results section.

c. Lines 453-476: This is Methods+Results mixed. It is also a justification of the effect not being an artifact of experimental design. Thus, this should go in Supplementary Info and the at Line 456 say: 'However, this was found not to be the case (Supplementary Info)". If indeed that is the conclusion - it was quite confusing.

d. Lines 344-345. 481-486 – delete – this is already in the Methods.

e. Lines 417-422: This is unclear. I think you have something here but it is not clear what. I think you are saying that the median of all estimated values was often close to the best fit value - suggesting even if you only did a single fit you might be in the right ballpark.... is this correct. If so, please rephrase.

f. Line 440-524: consider putting all rhos and their signif levels in a table - would shorten up the text (I know you took some of this out of tables on my request but now that you have shortened and refocused this, a single table with these results would be appropriate and help shorten the text).

3) SIGNIFICANCE TESTING. Line 440 onwards. The reporting of p-values in the text as % is non-standard and confusing. Please put these in as proportions – thus p=4.4% would be p=0.044. As you say in the Response to Reviewers, you should concentrate on the highly significant results. A p of 0.04 is so close to non-significant that given the number of tests, it should not be reported. Please consider removing the reporting of the rho and p for p > 0.01 from these, as there isn't sufficient confidence in these results (this will also make it easier to have a table for all these results, including only the highly significant ones). Also, it is not clear why the term “stochastically” is used. Please explain better in text.

4) CORRELATIONS BETWEEN PREDICTORS. Lines 506-524: This is a welcome discussion but I believe the magnitude of the correlation rather than the significance should be the focus here. It is possible to get very significant correlations that are not really very substantial (from many repeated measures or little variability). Thus, focus on magnitude. Since these are environmental factors I would argue that anything over rho = 0.6 (equivalent of an R2 of ~0.4) would be a fairly important correlation, but some would argue anything rho = 0.5 (R2 of 0.25) and above. Also, this would be better in a table showing the correlation matrix among variables. This could be in the Supplementary Info to shorten up the paper. Please note “Related Issue” is not an informative title.

5) DISCUSSION TOO BRIEF. Once all the methodological info is removed, the discussion is somewhat lacking. Please comment on what you have achieved – the discovery that the ratio is important. What does this mean going forward? How might ornithologists use this – do they need to, can they given the sample size restrictions? Certainly more discussion is needed.

a. Lines 587-598: This is only of importance for someone wanting to repeat your study to look at whether the ratio gives a biological signal. Instead, think of the limitations of avian ecologists - fewer data points , but likely more birds of a single species and fewer environmental factors. Given these considerations a better discussion paragraph could be made- the model has some relevance here - as greater sample sizes and perhaps more data points in species that take longer to develop (e.g. 20-40 days) could be better

6) COMPUTER PROCESSING TIME FOR FITTING. LINE 595 and response to comment 14. A data fit of 1 week of computer time per chick is outrageously slow and intense and would not appeal to many ornithologists. However, I believe that this is because you were validating the methods and your grid-search is fine-grained - to the 3rd significant figure. Commenting on this in the paper and indicating that the median ratio was fine and all values were within X units of the median suggests that a much coarser resolution could be used and that would give more realistic processing times - e.g if accuracy to 0.5 units was required how long would this take? For someone to want to fit these models something down to 1h per bird is more realistic (but still slow – considering it takes less than a second for the logistic curve). Perhaps, future research should explore results sensitivity to cutting the parameter search space and perhaps even explore different fitting methods (this could certainly be suggested).

A FEW OTHER SMALL POINTS:

Line 161 – Birds were marked: how? Using felt markers?

Line 165- it is more usual convention to call hatching date "Day 0"

Line 188 – NDVI - provide scale of resolution (1m2, 10m2, 1km2?)

Line 222: add (~5% of maximum mass) to indicate what 0.37g means in your case.

Line 348: Figure 3 only shows 1 bird so is not sufficient evidence. I would suggest that the fact that 94% of birds were normally-distributed then the underlying distribution is normal, and stating this is fine - the others could simply be random sampling errors.

Fig 4: these near diagonals actually look like pairs that are insensitive to the value of "a". This seems true of much of the data - it seems as though if b is low then a can take on any value, if b<6 then a <1, and otherwise as b increases a reduces.

Thank you again for the opportunity to review your interesting work.

Steve Oswald (sao10@psu.edu)

Reviewer #2: Review: Bertalanffy-Pütter models for avian growth

The article has changed considerably from the first version submitted.

Now the text is clearer and more objective and can be published.

I only recommend careful with the statement of line 205, because when stating

that the adjustment is made between the logarithm of the model and the logarithm

of the data, it is assumed that the error was multiplicative in the original Putter model (BP-model).

All suggestions from this reviewer were accepted and the paper was considerably improved.

The results are better discussed and therefore the paper has merit to be published.

I didn't check the layout question according to the journal's rules.

7. PLOS authors have the option to publish the peer review history of their article (what does this mean?). If published, this will include your full peer review and any attached files.

Reviewer #1: **Yes: **Stephen A. Oswald

Reviewer #2: **Yes: **Tales J. Fernandes

---

## [Author Response · Author response to Decision Letter 1]

13 Mar 2021

We revised the manuscript according the Reviewers comments.

Please find our respond in the Word file > RebuttalLetter_Revision2.

Kind regards, 

Manfred Kühleitner

---

## [Decision Letter · Decision Letter 2]

8 Apr 2021

Bertalanffy-Pütter models for avian growth

PONE-D-20-31677R2

Dear Dr. Kühleitner,

We’re pleased to inform you that your manuscript has been judged scientifically suitable for publication and will be formally accepted for publication once it meets all outstanding technical requirements.

Kind regards,

Dragan Pamucar

Academic Editor

PLOS ONE

Additional Editor Comments (optional):

Reviewers' comments:

Reviewer's Responses to Questions

**Comments to the Author**

1. If the authors have adequately addressed your comments raised in a previous round of review and you feel that this manuscript is now acceptable for publication, you may indicate that here to bypass the “Comments to the Author” section, enter your conflict of interest statement in the “Confidential to Editor” section, and submit your "Accept" recommendation.

Reviewer #1: All comments have been addressed

Reviewer #2: All comments have been addressed

2. Is the manuscript technically sound, and do the data support the conclusions?

Reviewer #1: Yes

Reviewer #2: Yes

3. Has the statistical analysis been performed appropriately and rigorously? 

Reviewer #1: Yes

Reviewer #2: Yes

4. Have the authors made all data underlying the findings in their manuscript fully available?

Reviewer #1: Yes

Reviewer #2: Yes

5. Is the manuscript presented in an intelligible fashion and written in standard English?

Reviewer #1: Yes

Reviewer #2: Yes

6. Review Comments to the Author

Reviewer #1: I thank the authors for their detailed responses and insightful revisions. I really enjoyed rereading this version of the manuscript and have no substantial comments that need addressing. I did catch a few typographical errors if the authors wish to fix these:

Line 250 - should the equal-to-or-less-than signs actually be equal-to-or-greater-than?

Line 270 - the period should be outside the parentheses

Line 298 - remove comma after "explored"

Line 371 - remove "=" sign

Line 441 - "test" should be "tests"

Line 463 - not that you have to mention it, but many semi-precocial and semi-altricial, ground-nesting seabird chicks make ideal subjects for this type of study and can be recaptured successfully until they fledge (e.g. Oswald et al. 2012 [72])

Line 488 - "allow to recognize" would be better as "permit recognition of"

Table 2: there are a few commas that should be decimal points in the table data

I thank you for the opportunity to review this very inspiring study.

Steve Oswald.

Reviewer #2: I would like to praise the hard work done by the authors in this paper and in corrections.

The methodology is now clearer and the results better discussed.

It is a work with potential to be cited because it collaborates with interesting aspects of nonlinear growth models.

7. PLOS authors have the option to publish the peer review history of their article (what does this mean?). If published, this will include your full peer review and any attached files.

Reviewer #1: **Yes: **Stephen A. Oswald

Reviewer #2: **Yes: **Tales J. Fernandes

---

## [Editor Report · Acceptance letter]

15 Apr 2021

PONE-D-20-31677R2 

Bertalanffy-Pütter models for avian growth 

Dear Dr. Kühleitner:

I'm pleased to inform you that your manuscript has been deemed suitable for publication in PLOS ONE. Congratulations! Your manuscript is now with our production department. 

Kind regards, 

on behalf of

Dr. Dragan Pamucar 

Academic Editor

PLOS ONE